# Tool Learning in the Wild:
# Empowering Language Models as Automatic Tool Agents

Anonymous

## Abstract

Augmenting large language models (LLMs) with external tools has emerged as a promising approach to extend their utility, enabling them to solve practical tasks. Previous methods manually parse tool documentation and create in-context demonstrations, transforming tools into structured formats for LLMs to use in their step-by-step reasoning. However, this manual process requires domain expertise and struggles to scale to large toolsets. Additionally, these methods rely heavily on ad-hoc inference technique or special tokens to integrate free-form LLM generation with tool-calling actions, limiting the LLM's flexibility in handling diverse tool specifications and integrating multiple tools.

In this work, we propose AUTOTOOLS, a framework that enables LLMs to automate the tool-use workflow. Specifically, the LLM automatically transforms tool documentation into callable functions, verifying syntax and runtime correctness. Then, the LLM integrates these functions into executable programs to solve practical tasks, flexibly grounding tool-use actions into its reasoning processes. Extensive experiments on existing and newly collected, more challenging benchmarks illustrate the superiority of our framework. Inspired by these promising results, we further investigate how to improve the expertise of LLMs, especially open-source LLMs with fewer parameters, within AUTOTOOLS. Thus, we propose the AUTOTOOLS-LEARNING approach, training the LLMs with three learning tasks on 34k instances of high-quality synthetic data, including documentation understanding, relevance learning and function programming. Fine-grained results validate the effectiveness of our overall training approach and each individual task. Our methods are an important step towards the use of LLMs for solving real-world tasks with external tools.

## Keywords

Large Language Models, Tool learning

## 1 Introduction

Large language models (LLMs) have shown promising capabilities such as in-context learning and real-world planning [1, 34, 39]. To further increase their utility, the tool learning task [19, 23] is proposed to augment LLMs with external tools, *e.g.,* a Weather App, enabling them to interact with the physical world [2, 18, 37], *e.g., look up the daily weather*. And most recent work further integrates

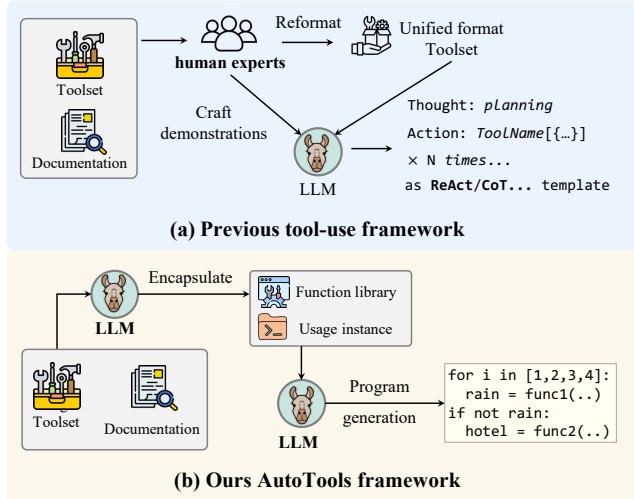

**(a) Previous tool-use framework**

**(b) Ours AutoTools framework**

**Figure 1: The comparison between conventional tool-use flow (a) and our proposed framework (b).**

tool-use LLMs with advanced inference techniques, such as ReAct [28, 33, 43] and tree-based search [20] or A-star algorithm [51], allowing them to server as agents to solve practical tasks.

**Augmenting LLM with tools.** Most previous work usually designs specific tool-use workflow for LLMs, integrating diverse tool-calling actions into the LLM generation process to solve practical tasks. Typically, they first pre-process toolset into a unified structure by manually understanding the development documentation of tools, such as web requests [28] or customized interfaces [23, 26], e.g., `translate[source] -> target`. Based on human expertise, developers craft elaborated instructions and few-shot demonstrations, instructing LLMs pre-defined usage templates and steering the generation format of LLMs. As shown in Figure 1, the LLM are guided to select useful tools in a step-by-step procedure, generate arguments for each selected tool in a pre-defined, customized format, and incorporate the response into subsequent action predictions.

However, these methods usually suffer from *two challenges* in realistic scenarios. *First*, it requires intensive expertise to effectively parse tool documentation and create valuable usage demonstrations, struggling to scale to large toolsets in practical applications. Consequently, LLMs show diminished performance when in-context examples are incomplete or missing, which potentially limits the scope of available tools to LLMs. *Second*, the tool-use workflow of LLM it is also ad-hoc to manually define the tool-use procedure and tool-calling format for LLM, showing limited generalization to diverse tool specifications. For example, ReAct [43] and ToolAlpaca [31] separately utilize each tool step-by-step, restricting their flexibility in integrating multiple tools dynamically in a once tool-calling action; The tool-calling template introduced in ToolLLM [20]

is far from that in ToolFormer [23], struggling to apply the Tool-LLM to the resource of ToolFormer. Therefore, a natural question is raised:

> Can we empower LLMs to automate tool-use flow and effectively manipulate diverse tools in the wild?

**LLMs as automated tool agents.** In this work, we propose a novel framework named AᴜᴛᴏTᴏᴏʟs, which diverges from previous work by enabling LLMs as agents to automate tool-use workflow. As shown in Figure 1(b), AᴜᴛᴏTᴏᴏʟs consists of two stages: (1) *Tool Encapsulation* and (2) *Tool Programming*.

In the *Tool Encapsulation* stage, AᴜᴛᴏTᴏᴏʟs automatically transforms the toolset into a list of well-structured, callable functions with generated demonstrations. Specifically, for each tool, the LLM is provided with its raw documentation and is induced to encapsulate it into a callable function. To verify the correctness, besides the syntax compilation, the LLM is stimulated to generate function-calling instances for each function to test the runtime correctness. Since relevant tools are the same resource and show strong input-output dependencies, we also propose an integration verification method, which enables LLM to integrate relevant functions to generate verification. The correct functions are augmented with its test instance as a usage demonstration and are gathered as a function library for the subsequent stage.

In the *Tool Programming* stage, the LLM is prompted to read the encapsulated functions and flexibly integrate them through a unified programming language (e.g., Python). Concretely, we first load the encapsulated functions to initialize an execution environment. Then, the LLM is equipped with the created function library and generates executable programs as a solution. The programs sequentially call a chain of functions, parse useful intermediates to resolve input-output dependencies among functions, and ultimately derive the final answer. By enabling the LLM as tool agents above, AᴜᴛᴏTᴏᴏʟs can benefit from the LLM's powerful abilities to transform abstract tool documentation into executable functions, yielding promising results in our pilot experiments.

Moreover, we further investigate how to improve the LLM's expertise within AᴜᴛᴏTᴏᴏʟs, especially for LLMs with fewer parameters. We propose AᴜᴛᴏTᴏᴏʟs-Lᴇᴀʀɴɪɴɢ, a multi-task learning approach that trains the LLM as an automated tool agent from synthetic datasets. We design three core learning tasks: (1) documentation understanding, where the LLM is trained to parse diverse tool documentation and generate structured functions; (2) relevance learning, where the LLM learns to select relevant tools based on a query and a candidate tool list; and (3) function learning, where we optimize the LLM to call in-context functions and solve practical queries. To enable this learning process, we filter and synthesize training data from large-scale public resources for each task, transforming it into a unified format. This enables us to collect high-quality examples without intensive human annotation.

**Experiments** We first evaluate our framework on two established benchmarks: RestBench [28] and ToolBench [20]. We also create a new benchmark named AᴜᴛᴏTᴏᴏʟs-Eᴠᴀʟ, including 224 tasks across 107 real-world tools, evaluating our framework in more challenging scenarios. AᴜᴛᴏTᴏᴏʟs-Eᴠᴀʟ diverges from the existing benchmarks by its more long-term planning tasks, complex tool documentation, and strong input-output dependencies among tools.

The results show that (1) LLMs like GPT-4 exhibit strong capabilities in understanding abstract tool documentation and generating callable functions; (2) AᴜᴛᴏTᴏᴏʟs substantially surpasses previous baselines with higher efficiency, and (3) AᴜᴛᴏTᴏᴏʟs-Lᴇᴀʀɴɪɴɢ further enhances the expertise of LLMs within AᴜᴛᴏTᴏᴏʟs.

**Contributions** Our contributions are as follows: (1) We propose AᴜᴛᴏTᴏᴏʟs, a framework combining tool encapsulation and tool programming, enabling LLMs to function as automated tool learners. (2) We introduce AᴜᴛᴏTᴏᴏʟs-Lᴇᴀʀɴɪɴɢ, a multi-task learning approach, and release 34k high-quality training data, further improving LLMs within AᴜᴛᴏTᴏᴏʟs. (3) Extensive experiments on both existing and newly collected datasets validate the superiority of our method. We will open-source AᴜᴛᴏTᴏᴏʟs for public use.

## 2 Related Work

**Tool learning with foundation models** Augmenting LLMs with real-world tools has been proven a promising method for enhancing their utility and enabling interactions with the physical world [2, 10, 18, 25]. To ground LLM with various tools, previous work first manually read development documentations of specific tools and process them into callable functions. In solving practical tasks, LLMs use tools by mimicking the handcrafted usage defined in their system prompts, typically generating parameters in structured formats to match predefined functions. Common practices include generating JSON (e.g., RestGPT [28], ToolLLM [20]), special tokens (e.g., ToolKenGPT [8]), or private function-calling messages (e.g., OpenAI's GPT). However, manually converting various tools into executable functions and carefully designing in-context examples for LLMs requires domain knowledge and experience, making it large to scale to massive toolsets. In this work, we investigate LLMs' expertise to automatically encapsulate tools into directly callable functions, thereby automating the above workflow.

**Programming-enhanced LLMs.** Recent work has shown the potential of using programming languages (PLs) to enhance the planning and reasoning capability of LLMs [11, 35, 41]. For example, previous work enables LLMs to generate a programmatic chain of thought to solve complex numeric reasoning tasks [3, 5], which exhibits remarkable performance. Compared with natural languages (NLs), recent studies also show that LLMs can generate Python code snippets as actions [33] or iteratively refine existing code [47]. In tool learning tasks, generating PLs benefits LLMs by integrating widely used packages such as TensorFlow [17] or Python pandas [33]. However, most existing work limits LLMs to using only well-processed functions, either manually simplified and encapsulated or frequently encountered during pre-training. In this work, our AᴜᴛᴏTᴏᴏʟs takes a further step by enabling LLMs to act as more automated tool-use agents, automatically generating directly callable functions grounded in corresponding tool documentation and creating demonstrations for in-contxt learning.

**Learning from external feedback.** Training LLMs using synthetic data is a widely-used method to improve their task-solving abilities and align them with following instructions [13, 16, 36]. Common practices for data synthesis include: (1) filtering data from large corpora like Common Crawl [32], (2) refining data quality through manual or automated processes [50], and (3) using LLMs to generate training data from scratch, as seen in approaches like

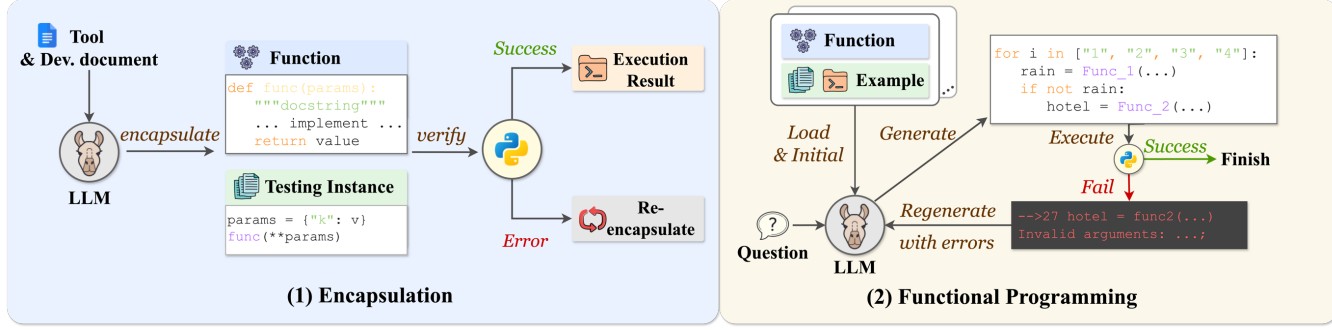

**Figure 2: An overview of the proposed framework AU⁠TO⁠TO⁠O⁠LS, in which the LLM (1) automatically encapsulates diverse tools into unified callable functions and (2) directly utilizes these functions through programming.**

SELF-instruct [34] and Magpie [40]. In the tool-learning task, previous work typically uses the third approach, specifically employing self-instruct [34] techniques to synthesize massive query-solution pairs [31, 42]. For example, ToolLLM [20] and Confucius [6] prompt LLMs to generate tool-use queries, then supplement them with chain-of-thought solutions that interleave tool names, tool arguments, and tool responses. Despite their advancements, generating data from scratch suffers from low diversity and uncontrollable quality [4, 13]. In contrast, our AU⁠TO⁠TO⁠O⁠LS-LE⁠A⁠RN⁠IN⁠G method synthesizes training data by filtering and reformatting various established datasets. Moreover, different from previous tool-use training, AU⁠TO⁠TO⁠O⁠LS-LE⁠A⁠RN⁠IN⁠G comprises three learning tasks, providing fine-grained supervision for tool understanding, query-tool relevance, and programmatic tool-use skills.

## 3 The Proposed Method: AU⁠TO⁠TO⁠O⁠LS

Our framework AU⁠TO⁠TO⁠O⁠LS is proposed to empower LLMs as automated tool agents to unified diverse tool-use specification and flexibly integrate them for task-solving, minimizing manual guidance. As illustrated in Figure 2, AU⁠TO⁠TO⁠O⁠LS consists of two core stages: (1) *tool encapsulation* and (2) *tool programming*. In the first stage, the LLM understand the development documentation $d$ of each tool $t$ and encapsulates it into a well-structured, callable function $f$. To verify the runtime correctness, we propose the *integration verification* method, which dynamically generates test instances to integrate relevant functions and check their execution result. The correct functions are augmented with test instances and gathered as a function library. In the second stage, instead of using original tools, the LLM directly integrates encapsulated functions by generating executable programs. Compared to other tool-use frameworks, AU⁠TO⁠TO⁠O⁠LS (1) automatically transforms abstract tool documentation into a callable function library and (2) allows the LLM to flexibly integrate multiple tools with different usage using a unified programming language.

## 3.1 Encapsulation: Tools $\overset{\text{LLM}}{\longrightarrow}$ Functions

We first introduce how to encapsulate a single tool into a well-structured function. As shown in Figure 2 (1), the LLM takes the tool documentation $d$ as input, which provides meta-information

in general natural language, such as tool arguments, functionality, optional access URLs and state code. The LLM aggregates the natural language descriptions of how to use the tool and grounds it to transform the abstract documentation into a directly callable function. Formally, this process can be represented as:

$$f = \mathcal{M}_\theta(t, d, \mathcal{I}_E),  \qquad (1)$$

where $\mathcal{I}_E$ represents the instruction for our encapsulation process. We use raw documentation as input $d$ since it can be easily obtained from official sources (e.g., RapidAPI platforms), minimizing the manual effort required by users in practical interactions. Since the LLM may hallucinate and miss necessary tool argument [52], we automatically compile the generated function into syntax tree [17] for syntax check. If any parameter name or type in the function signature does not exactly match the definitions in the tool documentation, the function is considered to fail the syntax check. If an error occurs, we repeat the Eq 3.1 for up to $m$ times.

## 3.2 Integration verification: Funcs $\overset{\text{verify}}{\longrightarrow}$ Func lib

Since syntax compilation fails to detect runtime errors in functions, it is crucial to design function-calling instances and verify the execution results. An intuitive approach to automate this process is utilizing the creative thinking abilities of LLMs [9, 24, 30], e.g., stimulating them to brainstorm test instances for each function individually. However, in large toolsets, tools within the same application often exhibit strong input-output dependencies. For example, a tool may require specific, private arguments derived from the output of another tool (e.g., retrieving movie credits relies on a unique ID as input). To address this, we propose integration verification, which identifies input-output dependencies between tools and verifies each encapsulated function by testing it in combination with its prerequisite functions.

Given a list of tools $T$, we sequentially encapsulate each tool $t_i$ into a function: $f_i = \mathcal{M}_\theta(t_i, d_i, \mathcal{I}_E)$, and initialize a cache $\mathcal{H}$ to store the correctly verified functions. The initial order can be a random permutation. To enable our integration verification, the LLM selects functions $\tilde{\mathcal{F}}$ relevant to $f_i$ from the cache $\mathcal{H}$:

$$\tilde{\mathcal{F}} = \mathcal{M}_\theta(f_i, \mathcal{F}, \mathcal{I}_{\text{Rel}}).  \qquad (2)$$

**Figure 3: Details for our integration verification (Section 3.2).**

Here, $\mathcal{I}_{\text{Rel}}$ is the instruction for relevance selection and the functions in $\tilde{\mathcal{F}}$ can be empty if $\mathcal{H}$ is empty or the $f_i$ has no private argument requirements. Then, the LLM obtain the necessary input parameters for function $f_i$ using $\tilde{\mathcal{F}}$ and generate a test instance $e_i$:

$$e_i = \mathcal{M}_\theta(f_i, \tilde{\mathcal{F}}, \mathcal{I}_{\text{E}}) \tag{3}$$

If function $f_i$ is verified as correct, it is moved from $\mathcal{T}$ to $\mathcal{H}$. We repeat this process for each tool in $\mathcal{T}$ along the initial order. This iterative traversal continues until $\mathcal{T}$ is empty or up to the maximum traversal times. We augment the correct functions from cache $\mathcal{H}$ with its instance as an in-context demonstration.

## 3.3 Tool Programming: LLM $\xrightarrow{\text{Func lib}}$ solution

In the tool programming stage, AutoTools allows LLM to seamlessly integrate the executable functions for task-solving instead of using abstract tool documentation as previous work. Using the pre-encapsulated and verified functions can reduce potential misunderstanding towards abstract tool documentation. Besides, different from using customized output templates or special token, the AutoTools allows LLMs to directly manipulate multiple functions using an unified programming language. The LLM flexibly integrates function-calling actions into its generation by generating executable programs.

Given a practical query $q$, the LLM is equipped with its generated function library $\mathcal{F} = \{(f_i, c_i, r_i) \mid i \leq |\mathcal{F}|\}$. Here, $f_i$ is a callable function with a well-structured docstring, $c_i$ provides a default usage example, and $r_i$ specifies the expected execution result type. We first load these functions into the execution environment $E$ and initialize a session to interact with the LLM. We instruct the LLM $\mathcal{M}_\theta$ to generate an executable program as a solution $s$. The program $s$ sequentially calls pre-encapsulated functions, parses execution results for further use, and simplifies the task-solving process with concise programmatic control flow statements like for-loop. The final result $r$ is derived by executing the generated program, which returns either the correct result or error messages. During the interaction, our execution session caches variables defined by the LLM for reuse in subsequent programs. The session terminates when the LLM outputs Finish. Formally, this process can be formulated as:

$$s_j = \mathcal{M}_\theta(q, \mathcal{F}, \mathcal{I}_{\text{P}}, \{(s_{<j}, r_{<j})\}) \tag{4}$$

Here, the $\mathcal{I}_{\text{P}}$ indicates a concise instruction for program generation operation, which is provided in Appendix A.5. We set the maximum interaction number as $m$.

## 4 Learning with AutoTools

Our AutoTools empowers the LLM as a tool agent, which benefits from the LLM's powerful abilities to transform abstract tool documentation into executable functions. In our pilot experiment, LLMs like GPT-4, when equipped with AutoTools, show substantial improvements. Motivated by these promising results, we further investigate how to improve the LLM's expertise within AutoTools, especially for open-source LLMs with fewer parameters. To achieve this, we propose AutoTools-Learning, which consists of three learning tasks in which the LLM $\mathcal{M}_\theta$ learns to encapsulate tools into functions and effectively utilize these functions. In this section, we introduce the objective of each learning task and detail how to synthesize the training data to enable this learning process.

### 4.1 Learning Tasks and Objectives

We propose the following three learning tasks.

**Tool understanding** In this task, we train the LLM to comprehend complex tool documentation, which provides raw information on how to invoke the tool. Formally, given a tool $t$, the LLM is trained to generate a well-structured function $f$ based on the tool documentation $d$. It can be formulated as:

$$\mathcal{L}_{\text{Und}} = -\log P_\theta(f|\mathcal{I}_{\text{E}}, t, d) \tag{5}$$

The $\mathcal{I}_{\text{E}}$ indicates the instruction for encapsulation operation mentioned in Eq 3.1. The function $f$ encapsulates detailed tool-calling information from $d$, such as web request headers, base URLs, and exception handling. Additionally, it includes a standard function signature and a docstring to demonstrate its arguments and expected execution results.

**Relevance learning** Since solving a user's query in practical scenarios typically involves multiple tools, we design a relevance learning task, teaching the LLM how to select the most useful tools from a candidate toolset. Given a list of tools $\mathcal{F}$ with detailed docstring, we formulate this learning process as a generative task, where the LLM is trained to autoregressively generate the identifiers (i.e., names) of relevant functions. Assume the $\tilde{\mathcal{F}} = \tilde{f}_i \mid i \leq |\tilde{\mathcal{F}}|$ is the ground truth function relevant to a query $q$, we concatenate the identifiers of relevant functions as $y = \tilde{f}_1 \oplus \tilde{f}_2 \oplus \cdots \oplus \tilde{f}_{|\tilde{\mathcal{F}}|}$, Then, we apply the standard language modeling loss:

$$\mathcal{L}_{\text{Rel}} = -\sum_{t=1}^{|y|} \log P_\theta(y_t|y_{(<t)}, \mathcal{I}_{\text{Rel}}, q, \mathcal{F}), \tag{6}$$

where $\mathcal{I}_{\text{Rel}}$ is the task instruction for relevance selection. This listwise selection manner allows the LLM to compare multiple similar tools and determine the query-tools relevance during the token-by-token prediction process [29].

**Function learning** Our function learning grounds the LLM in the practical task-solving process with the assistance of the provided functions. Starting with a user query $q$, we establish a multi-turn session between the LLM and the execution environment. Specifically, the LLM is trained to generate programs that call various functions $\mathcal{F} = \{f_i|i \geq |\mathcal{F}|\}$ provided in-context, receiving execution results

**Table 1: The data scale of our synthetic training dataset and detailed average statistics *per example.***

| Statistic | |
|---|---|
| # The data scale | 34,183 |
| # The average length of input | 664.80 |
| # The average length of output | 264.40 |
| # The average number of candidate tools | 8.23 |
| # The average turn of interaction | 2.66 |

from the environment to determine its next step. Assuming $h_j$ is the interaction history up to turn $j$, the optimization objective for $j$th turn can be formulated as:

$$\mathcal{L}_{\text{Func}} = -\log P_\theta(c_j | \mathcal{I}_{\text{Func}}, q, \mathcal{F}, \{(c_{<j}, r_{<j})\}) \tag{7}$$

By generating executable programming language, the LLM can inherently manipulate tools using built-in control flow statements, (e.g., for-loops and if-else statements), and store useful intermediates for subsequent reuse.

## 4.2 Training Data Synthesis

**For the tool understanding task**, we first collect a large number of tools from the ToolBench [20] dataset. Each tool is originally crawled from the RapidAPI platform and has been manually supplemented with its callable function, making it inherently similar to the setting of this learning task. **For the relevance learning task,** we gather data from various tool retrievals datasets, such as COLT [21] and APIGen [15], where each example consists of a query, a list of candidate tools, and the target tools. We first transform these tools into a unified function through our encapsulation operation in Section 3.1. Then, we unify this data into a listwise selection format similar to RankGPT [29]. **For the function learning task**, we collect step-level task-solving trajectories from existing tool-use datasets. We then use a powerful LLM, i.e., GPT-4o, to generate program solutions by referencing the originally provided ground truth, aligning the data with our function learning setting.

To ensure data quality, we apply strict filtering strategies, such as removing examples with empty tool responses, unsolvable queries, or incorrect tool-calling parameters. Ultimately, we collect 7,243/12,251/14,689 examples for the three tasks, respectively. We also reformat these datasets into a unified interactive format, similar to prior work [45, 48]. Each formatted example begins with a system instruction describing the task and initial input, followed by interactions between two roles: the user and the LLM, or the LLM and the execution environment. We report the statistics of the final training data in Table 1 with additional details provided in Appendix A.1. Our overall optimization combines the three tasks, enhancing the LLM's expertise in AUTOTOOLS through a multi-task learning approach.

## 5 Dataset and Evaluation Setup

### 5.1 Dataset

**Existing Datasets** We first conduct experiments on two widely used benchmarks: RestBench [28] and ToolBench. RestBench consists of two subsets: (1) TMDB, a high-quality, human-annotated dataset comprising 54 movie-related tools, and (2) Spotify, a dataset

**Table 2: The comparison between our newly collected benchmark AUTOTOOLS-EVAL with existing benchmarks (test set).**

| Dataset | # Task | # Tool | Path len. | Doc len. |
|---|---|---|---|---|
| AUTOTOOLS-EVAL | 224 | 7.31 | 107 | 552.92 |
| RestBench [28] | 157 | 2.36 | 94 | 716.69 |
| ToolBench [20] | 600 | 2.56 | 1806 | 159.47 |
| ToolBench-sam [38] | 895 | 5.35 | 232 | 66.98 |
| APIbank [12] | 272 | 1.99 | 101 | 75.85 |
| ToolEyes [44] | 382 | 2.00 | 568 | 72.06 |

containing 40 music-related tools. ToolBench includes various practical tasks across diverse scenarios. Each tool in the RestBench is paired with a lengthy documentation, making it inherently appropriate to benchmark the tool understanding capability of LLMs.

**A new benchmark – AUTOTOOLS-EVAL** As shown in Table 2, to the best of our knowledge, no existing benchmarks containing complex tools with complex tool documentation while involving long-term planning tool-use tasks. Therefore, we build a new test set named AUTOTOOLS-EVAL to fill this gap. We first collect 107 tools with long documentation across 4 real-world domains, *e.g.,* Weather and Game, from 16k public tools of the ToolBench [20] dataset. Then, we invite 7 well-trained experts working on NLP research to provide solutions for 224 complex task. Each task requires long-term reasoning and at least 7 times tool-callings. AUTOTOOLS-EVAL also diverges from existing benchmarks by its strong interconnection among the tools (the arguments of subsequent tools can only be extracted from the response of previous tools) and stability (the task solution is not time-varying). We provide more details of AUTOTOOLS-EVAL in Appendix A.3.

### 5.2 Evaluation metrics

Following previous work [20, 27, 28, 42], we evaluate the task-solving performance of our AUTOTOOLS and tool-use framework using the following metrics. For RestBench, we use three evaluation metrics including: (1) Success Rate (**Success%**), which measures whether all the required tools (ground truth tools) are correctly called to solve the task [28, 42]; (2) Correct Path Rate (**Path%**), which calculates the proportion of ground truth tools in model-generated tool callings; (3) Correct Tool Precision (**Prec%**), which calculates the precision score between the model-generated tool callings and ground truth tool sequence. For ToolBench, we also use the **Pass Rate** as a metric following its official evaluation script, which evaluates whether the model successfully completes a solvable task or try necessary tools but give up a unsolvable task. Additionally, to evaluate the LLMs' performance in encapsulating tools, we use *the number of correctly encapsulated tools* as a evaluation metric.

### 5.3 Baselines

We mainly compare our AUTOTOOLS with the well-known baselines, including: (1) ReAct [43], which prompts LLM to generate the chain-of-thought and actions in an interleaved manner; (2) CodeAct [33], which prompts LLM to iteratively generate code snippets as actions to call manually demonstrated tools. (3) ToolLLM-DFSDT[20],

which enhances LLMs with the Depth First Search-based Decision Tree (DFSDT) to select tools to solve a task; (4) RestGPT [28], which includes a coarse-to-fine planning module and a tool executor; (5) ConAgents [27], which enables the cooperation of three specialized LLMs to solve complex tasks. For further comparison, We also establish two baselines, *i.e.,* ReAct@3 and ToolLLM@3, which are up to three times runs of their vanilla method (ReAct or ToolLLM) until the input task is successfully completed.

We follow the official implementation for each baseline method, providing LLMs with well-demonstrated usage and detailed instruction in their system prompt to master the tool usage. And we mark the baseline which relies on OpenAI's official function-calling technique[1] with $^\dagger$. Besides, following previous work [20, 31], each evaluation task is officially paired with a candidate toolset about 20 tools as the input for all the methods. Each toolset contains the required tools (ground truth) and randomly sampled tools.

## 6 Experimental Results

In this section, we conduct extensive experiments to answer the following research questions:

**RQ1:** Can LLMs understand documentation and automatically encapsulate functions?

**RQ2:** To what extent does our AUTOTOOLS improve the performance of LLMs?

**RQ3:** To what extent does AUTOTOOLS-LEARNING enhance the ability of the LLM in AUTOTOOLS?

**RQ4:** Is AUTOTOOLS more efficient for task-solving compared to existing approaches?

### 6.1 RQ1 – Performance on tool encapsulation.

We first investigate the LLMs' expertise in tool encapsulation. For comprehensive evaluation, we conduct experiments on a series of widely used LLMs, including: (1) *GPT-4-turbo*, (2) *GPT-3.5-turbo-16k*, (3) *Mixtral-8x7B*, (4) *Mistral-7B-instruct*, and (5) *Llama-3-8B-instruct*. Specifically, we report the number of correctly encapsulated functions as the evaluation metric. The sampling number $n$ is set to 3, and the maximum traversal number is set to 4.

**Experiment results** Table 3 shows the number of correctly encapsulated tools. We observe that powerful LLMs, such as GPT-4, can encapsulate almost 90%~95% tools into well-structured functions, exhibiting remarkable performance. Besides, the open-source model Mixtral-8x7B correctly encapsulate 82.5% to 88.2% tools into functions, achieving promising results. These findings illustrate that LLMs are capable of understanding tool documentation and generating callable functions. A potential explanation is that LLMs have been trained on large-scale web corpora that include diverse code and API documentation resources, allowing them to acquire the necessary understanding skills during the pre-training stage.

**Ablation study.** In our experiment, we verify the correctness of the encapsulated functions via syntax compilation (Section 3.1) and integration verification (Section 3.2). We compare our vanilla method with two ablative variants: (1) *w/o syntax*, which removes the syntax compilation, and (2) *w/o integrate*, which sequentially encapsulates each tool without integrating relevant tools. As shown in Table 3, in terms of the number of correct encapsulation numbers,

[1]https://platform.openai.com/docs/guides/function-calling

**Table 3: The number of *correctly encapsulated tools* using our vanilla method and two variants on benchmarks (test set). Ours-Eval indicates our collected dataset AUTOTOOLS-EVAL.**

| Backbone | TMDB | Spotify | Ours-Eval | ToolBench |
|---|---|---|---|---|
| **Totally** | **54** | **40** | **107** | **3211** |
| gpt-4-turbo | 54 | 38 | 102 | 3071 |
| mixtral-8x7B-inst. | 48 | 35 | 95 | 2793 |
| mistral-7B-inst. | 45 | 32 | 92 | 2647 |
| Llama-3-8B-inst. | 42 | 32 | 90 | 2582 |
| gpt-3.5-turbo-16k | 54 | 38 | 98 | 2990 |
| - *w/o syntax* | $50_{\downarrow 4}$ | $35_{\downarrow 3}$ | $91_{\downarrow 7}$ | $2497_{\downarrow 493}$ |
| - *w/o integrate* | $47_{\downarrow 7}$ | $17_{\downarrow 21}$ | $87_{\downarrow 11}$ | $2655_{\downarrow 335}$ |

we observe 3-7 point decreases for *w/o syntax*, which indicates that the LLMs may fail to generate a correct program at one pass. We further analyze the error cases and find that LLMs may hallucinate by generating non-self-contained functions that depend on undefined or randomly fabricated variables. Besides, we find a substantial decrease between our vanilla method and the *w/o integrate* variant. These results demonstrate the necessity of optimizing the integration of the function with strong input-output dependence.

### 6.2 RQ2 – Overall Performance

The results of RQ1 demonstrate that LLMs show promising capability in automatically encapsulating tools into callable functions. In RQ2, we further evaluate the LLMs' expertise in manipulating pre-encapsulated functions to solve practical tasks within the proposed AUTOTOOLS. We set the maximum interaction turns to $m = 5$ (Section 3.3) and conduct comprehensive experiments on LLMs with varying parameter scales.

**Results on existing benchmarks** As shown in Table 4, the LLM, when equipped with our framework, surpasses all the baselines on the RestBench and ToolBench benchmark across all metrics. For example, AUTOTOOLS achieves 89.00% in success rate metrics on the TMDB (RestBench) dataset, which substantially improves both the commonly used ReAct and the more advanced ToolLLM. Table 5 further illustrates that our framework achieves the best performance with various backbone LLMs, *i.e.,* the Mistral-8x7B and GPT-4. These results indicate that our framework effectively enables LLM to master executable functions and effectively integrate them to solve complex tasks. The performance of two runs is tested using a two-tailed paired t-test where no significant difference is found ($p > 0.05$), showing the stability of our method.

**Results on AUTOTOOLS-EVAL** Table 4 presents the results on our AUTOTOOLS-EVAL benchmark. We find that our AUTOTOOLS-EVAL poses a substantial challenge for previous baselines, with the best performance only achieving a 44.70% success rate using GPT-3.5 as the backbone. In contrast, our method improves the success rate to 60.21%, representing a 15.51 point increase. This improvement is attributed to our AUTOTOOLS framework, which grounds LLMs with diverse tools by enabling them to integrate pre-encapsulated functions through programming. The LLM generates

**Table 4: Experiment results on three datasets with gpt-3.5-turbo as the backbone. The Path Rate, Precision, and Success% indicate *Correct Path Rate, Correct Path Precision,* and *Successful Rate* metrics. \*The Precision of ToolLLM is substantially lower than other baselines since it employs a DFS search algorithm to repeatedly call incorrectness tools instead of stopping.**

| Method | TMDB (RestBench) | | | Spotify (RestBench) | | | AUTOTOOLS-EVAL | | | ToolBench |
|---|---|---|---|---|---|---|---|---|---|---|
| | Success% | Path Rate | Precision | Success% | Path Rate | Precision | Success% | Path Rate | Precision | Pass Rate |
| *gpt-3.5-turbo-16k* | | | | | | | | | | |
| ReAct[†] [43] | 61.00 | 77.13 | 52.30 | 50.88 | 74.64 | 44.79 | 22.76 | 60.75 | 68.03 | 39.39 |
| CodeAct [27] | 63.00 | 80.91 | 83.72 | 54.30 | 76.64 | 79.81 | 27.82 | 57.93 | 66.23 | - |
| ToolLLM[†] [20] | 72.00 | 78.29 | 49.41 | 61.40 | 82.82 | 25.33* | 42.14 | 71.02 | 65.24 | 66.39 |
| RestGPT [28] | 65.00 | 77.49 | 80.15 | 64.91 | 73.94 | 88.71 | 26.83 | 40.95 | 62.21 | 63.88 |
| ConAgents [27] | 76.00 | 78.29 | 82.31 | 63.16 | 78.21 | 82.71 | 60.21 | 78.31 | 72.45 | 69.84 |
| ReAct@3[†] | 70.00 | 80.96 | 48.01 | 59.65 | 81.80 | 30.48 | 28.35 | 66.66 | 66.21 | 66.12 |
| ToolLLM@3[†] | 74.00 | 83.29 | 45.41 | 66.67 | **83.41** | 23.73 | 44.70 | 73.85 | 60.77 | 68.77 |
| **AUTOTOOLS (ours)** | **89.00** | **84.71** | **83.87** | **78.95** | 78.54 | **91.46** | 60.21 | 78.31 | 72.45 | **75.21** |

**Table 5: Experiment results on more widely-used LLMs to validate the effectiveness of our AUTOTOOLS.**

| Method | TMDB | | AUTOTOOLS-EVAL | |
|---|---|---|---|---|
| | Success% | Path Rate | Success% | Path Rate |
| *gpt-4-turbo* | | | | |
| ReAct[†] | 77.00 | 86.05 | 25.99 | 65.98 |
| ReAct@3[†] | 80.00 | 89.21 | 30.98 | 67.55 |
| ToolLLM@3[†] | 82.00 | 90.62 | 50.46 | 76.73 |
| **Ours** | **94.00** | **92.68** | **65.74** | **83.54** |
| *mixtral-8x7B-instruct* | | | | |
| ReAct | 24.74 | 73.34 | 10.53 | 41.37 |
| ReAct@3 | 37.88 | 76.85 | 18.95 | 52.40 |
| ToolLLM@3 | 45.00 | 74.40 | 22.54 | 51.85 |
| **Ours** | **58.00** | **78.17** | **29.87** | **59.14** |
| *Llama3-8B* | | | | |
| ReAct@3 | 15.00 | **56.25** | 0.00 | 25.59 |
| ToolLLM@3 | 15.15 | 50.51 | 1.74 | 31.04 |
| **Ours** | **18.00** | 54.31 | **5.36** | **36.24** |
| *mistral-7B-instruct* | | | | |
| ReAct@3 | 12.00 | 57.10 | 4.35 | 35.95 |
| ToolLLM@3 | 18.00 | **60.14** | 5.09 | 37.32 |
| **Ours** | **23.00** | 59.62 | **10.71** | **40.31** |

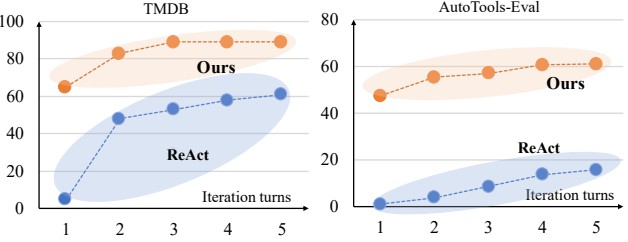

**Figure 4: The step (turn) level performance evaluation.**

**Table 6: Ablation study of our AUTOTOOLS-LEARNING. We investigate the effectiveness of each learning task.**

| Method | TMDB | | AUTOTOOLS-EVAL | |
|---|---|---|---|---|
| | Success% | Path% | Success% | Path% |
| *mistral-7B-instruct* | | | | |
| **Ours** (vanilla) | 23.00 | 59.62 | 10.71 | 40.31 |
| **Ours** (trained) | **29.00** | **64.10** | **16.52** | **44.56** |
| - w/o $\mathcal{L}_{Und}$ | $26.00_{\downarrow 3.0}$ | $62.13_{\downarrow 2.0}$ | $14.29_{\downarrow 2.2}$ | $42.35_{\downarrow 2.2}$ |
| - w/o $\mathcal{L}_{Rel}$ | $26.00_{\downarrow 3.0}$ | $61.04_{\downarrow 3.1}$ | $15.18_{\downarrow 1.3}$ | $41.37_{\downarrow 3.2}$ |
| - w/o $\mathcal{L}_{Func}$ | $25.00_{\downarrow 4.0}$ | $62.76_{\downarrow 1.3}$ | $13.39_{\downarrow 2.1}$ | $42.52_{\downarrow 2.0}$ |

directly executable programs, flexibly integrating multiple tool-calling actions into the long-term reasoning process.

**Analysis on interaction turns.** We further investigate the LLM's performance as the maximum interaction turns $m$ vary from 1 to 5, with the results shown on two datasets in Figure 4. On the AUTOTOOLS-EVAL dataset, we observe an increasing success rate as $m$ shifts from 1 to 4, followed by a relatively stable trend as $m$ increases from 4 to 5. These results indicate that the LLM can correctly call the required tools and revise errors in approximately three steps. Given that each task in AUTOTOOLS-EVAL requires an average of 7.31 tool calls (see Table 2), our AUTOTOOLS enables the LLM to generate executable programs that directly integrate multiple functions. A similar trend is observed in the TMDB dataset,

where the LLM completes tasks in just 2 turns, compared to an average task path length of 2.36 in TMDB.

## 6.3 RQ3 – Further improvement

Our AUTOTOOLS-LEARNING is proposed to further improve the LLM's expertise within AUTOTOOLS, which trains open-source LLMs using synthetic examples through the multi-task learning. We employ the DeepSpeed ZeRO-3 strategy [22], with a learning rate of $2e^{-5}$ and 3 training epochs on 8 NVIDIA A100-PCIE-80GB GPUs. We compare the performance of AUTOTOOLS with both trained and vanilla (i.e., out-of-the-box) LLMs. Table 6 presents the experiment results. We obverse that our AUTOTOOLS-LEARNING substantially

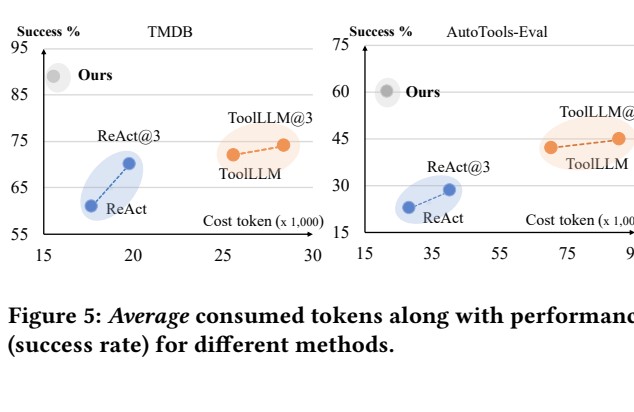

**Figure 5:** *Average* consumed tokens along with performance (success rate) for different methods.

improves overall performance of the Mistral-7B. For example, it pushes the success rate to 16.52 in the AUTOTOOLS-EVAL dataset.

To further evaluate the effectiveness of the three learning tasks in the AUTOTOOLS-LEARNING, we also conduct a fine-grained ablation study, removing each task in turn and training the LLM with only the remaining two tasks.

**w/o** $\mathcal{L}_{\mathbf{Und}}$. We remove the *document understanding* task formated in Eq 5. As illustrated in Table 6, the success rate decreases by 3.00 points in the TMDB dataset and by 1.3 points in the AUTOTOOLS-EVAL dataset. These results highlight the importance of the document understanding task in enhancing overall performance.

**w/o** $\mathcal{L}_{\mathbf{Rel}}$. We remove the *relevance learning* task defined in Eq 6. A decrease in the correct path rate metric is observed across both datasets, validating the necessity of learning the relevance between the query and candidate tools.

**w/o** $\mathcal{L}_{\mathbf{Func}}$ We remove the *function learning* task formulated in Section 7. The success rate decreases by 4.00 points in the TMDB dataset (17.39% relative improvement) and by 2.1 points in the AUTOTOOLS-EVAL dataset (19.61% relative improvement). Besides, removing this task has the most pronounced impact compared to the removal of the document understanding or relevance learning tasks. This finding suggests that function learning is more fundamental to our AUTOTOOLS-LEARNING, and training the LLM with this task is crucial to optimize its performance within the AUTOTOOLS.

### 6.4 RQ4 – Efficiency analysis

We further analyze the **efficiency** of our framework compared to strong baselines in the task-solving process. Figure 5 shows the token consumption alongside the performance results for a more intuitive comparison. We show their **consumed token** along with their performance results on to explain more intuitively. We observe that, despite achieving better performance, our framework consumes fewer tokens compared to all baselines. The reason is that our framework allows the LLM to flexibly integrate well-encapsulated functions and transform multi-step tool-callings into a concise, structured program. We also compute the token consumption for our encapsulation process in Table 8. Given comprehensive tool documentation, we find that GPT-3.5-turbo-16k only consumes 2703 tokens to encapsulate a tool into a callable function with usage examples. These encapsulated functions can be cached and loaded for subsequent reuse. More details can be found in A.2.

**Table 7: The statistics of the error of our framework.**

| Error analysis | Percent% |
|---|---|
| # 1. *Selection error*: confuse similar tools or only select part of required tools | 44.0% |
| # 2. *Arguments error*: make up non-exist variables | 25.2% |
| # 3. *Parse Error*: hallucinate the structure and type of function return value | 30.8% |

## 7 Discussion

**Statistics of error cases.** To further evaluate the potential strengths and weaknesses of our method, we analyze the types of failure cases, categorizing them into three groups, as shown in Table 7. Most errors stem from selecting incorrect functions or mismatching the expected return value types of similar functions. Thus, we conducted an additional experiment under the same conditions as Table 4, except that we reduce the number of candidate tools for each test query from 20 to 10. We observe a 2-3 point improvement in performance across the RestBench, AUTOTOOLS-EVAL, and Tool-Bench datasets. Thus, we believe that a solution to mitigate the errors identified in Table 7 is to filter out irrelevant functions (e.g., using embedding or retrieval models) as proposed in [21], thereby reducing noise for tool-use LLMs.

**Case study.** Besides automatic evaluation in our experiment, we also conduct case studies and human evaluation for a comprehensive evaluation. The concrete examples and results are shown in Appendix A.4 for an intuitive explanation.

## 8 Conclusions

We presented AUTOTOOLS, a framework that enables LLMs to act as automated tool learners, automating the tool-use workflow. Within AUTOTOOLS, the LLM first transforms tool documentation into callable functions, verifying both syntax and runtime correctness. It then integrates these functions into executable programs, flexibly grounding tool-use actions within its reasoning processes to solve practical tasks. AUTOTOOLS addresses two key challenges in existing tool learning methods: (1) reliance on intensive human expertise to process diverse and complex tool documentation into structured formats with in-context examples, and (2) the limitations of handcrafted, ad-hoc control flows to integrate LLM generation with diverse tool-calling actions. Extensive experiments on existing datasets and a newly created challenging benchmark demonstrate the effectiveness of our framework. Inspired by the promising performance of AUTOTOOLS, we further propose the AUTOTOOLS-LEARNING, which enhances LLM capabilities, particularly for open-source LLMs with fewer parameters. We expect future research to integrate our framework into vision foundation models, developing multi-modal agents for real-world task-solving.

### Ethical Use of Data and Informed Consent

We followed ethical standards, using publicly accessible tools and benchmarks to ensure transparency, reproducibility, and fairness in our research. we ensured that our methods are free from harm or deception and do not produce toxic outputs.

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

# A  Appendix

## Ethical Use of Data and Informed Consent

The research conducted in this paper aims at the development of empowering large language models (LLMs) as automated tool learners. It enables LLMs to transform abstract tool documentation into executable function libraries and to flexibly integrate functions through programming to solve practical tasks. In the process of conducting this research, we have adhered to ethical standards to ensure the integrity and validity of our work. All tools used in this study were obtained from publicly accessible platforms or widely-used benchmarks, ensuring transparency and reproducibility in our experiments and minimizing potential bias and promotes fairness.

We have made an effort to ensure that our research does not harm individuals or groups, nor does it involve any form of deception or potential misuse of information. The tools used in this research do not pose any harm, and there is no malicious behavior associated with the LLMs or the tools. Additionally, we have ensured that the LLMs do not produce harmful or toxic outputs. Our code, prompts, and datasets will also be open-sourced to facilitate further research, making them available after the anonymization period.

## A.1  Training Data Synthetic

Our AᴜᴛᴏTᴏᴏʟs-Lᴇᴀʀɴɪɴɢ trains the LLM using a synthetic dataset through a multi-task learning approach, which includes three key tasks: tool understanding, relevance learning, and function learning. The AᴜᴛᴏTᴏᴏʟs-Lᴇᴀʀɴɪɴɢ synthesizes training data by reformatting established datasets into an interactive task-solving format, simulating interactions between the user and the LLM, or the LLM and the execution environment. Below, we detail the data resources for each task, respectively.

**Data synthetic for the tool understanding task.** We first collect a large number of tools (16k) from the ToolBench [20] dataset. Each tool is originally crawled from the RapidAPI platform and has been manually supplemented with its callable function, making it inherently similar to the setting of our learning task. The input for each training example in this task is the tool's development documentation, while the output is a well-structured Python function pre-created by ToolBench. The tool documentation includes an abstract description of how to invoke the tools. The generated functions are directly callable and executable.

**Data synthetic for the tool understanding task.** We gather data from various tool retrieval datasets, including (1) ToolACE [14], (2) ToolBench [20], (3) APIGen [15], (4) Confucius [6], and (5) ToolAlpaca [31]. Each example consists of a query, a list of candidate tools, and the target tools. We first transform the tools into a unified function using the encapsuluation operation in Section 3.1 and we unify this data into a listwise selection format, similar to RankGPT [29] and RankRAG [46]. In this task, the input of each training example is the concatenation of the query and the tools, while the output is the unique ID of the ground truth tool. Here, the unique ID for each tool specifically indicates the tool name.

**Data synthetic for the function learning task.** We collect step-level task-solving trajectories from existing tool-use datasets, including (1) ToolACE [14], (2) ToolBench [20], and (3) APIGen [15].

We selected these datasets because they have provided large-scale training sets rather than just test sets. Each example in this task consists of a practical query (e.g., `fetch the past month's Daily 4 lottery results?`), a list of relevant tools, and the step-level solution. The solution involves tool-use actions, such as selecting relevant tools (e.g., `Daily_4_History_API`), specifying parameters (e.g., `start=2022-05-20, end=2022-06-20`), and receiving the response. We filter out low-quality examples that contain unsolvable queries or empty tool responses. Then, we use a powerful LLM (GPT-4o) to generate program solutions based on the originally annotated solution, reformatting the customized tool-use actions into unified programs. In this process, their pre-annotated solutions are used as references to ensure the correctness of the reformatted data. Besides, if a reformatted example contains syntax errors or tool-calling parameters that differ from its pre-annotated solution, it is discarded.

In total, we collect 7,243/12,251/14,689 examples for the above three tasks, respectively. We also reformat these datasets into a unified interactive format, similar to previous work [31, 45]. Each formatted example begins with a system instruction describing the task and initial input, followed by interactions between two roles: the user and the LLM, or the LLM and the execution environment. Our overall optimization involves combining the three tasks to optimize the LLM's expertise in AUTOTOOLS through a multi-task learning approach.

## A.2 More Experiment Details

*The tool encapsulation.* In our experiments, we evaluate our encapsulation method for four datasets, *i.e.,* RestBench-TMDB, RestBench-Spotify, AUTOTOOLS-EVAL, and ToolBench, respectively. We provide the cost statistic for this process in Table 8.

**Table 8: Detailed statistic of our tool encapsulation.**

| Statistic | |
|---|---|
| Maximum number of iterations per tool | 4 |
| Runtime iterations during the experiment | 3 |
| Avg. encapsulation attempts per tool | 2.04 |
| Avg. token consumption per tool | 2703 |

*The runtime consistency of our experiment.* Since the non-deterministic generation of LLMs by nature, we further explore the consistency and stability of our framework. We repeat our method (**ours**) with the same setting as Table 4 in RestBench. The statistical significance of differences observed between the performance of two runs is tested using a two-tailed paired t-test. We find no significant difference between the results of two randomly conducted experiments ($p > 0.05$).

*Human evaluation.* ollowing previous work [20, 28], we conduct a human evaluation on two metrics, including: (1) Executability (Exec): whether multiple tools are invoked in a correct logical order to complete the task; (2) Tool utilization (Uility): whether the model can observe the relevant values from lengthy execution results and incorporate them to predict the next action. We invite three

**Table 9: The human evaluation on three datasets for executability and utility. Scores are on a scale of 1–3.**

| | ReAct | CodeAct | ToolLLM@3 | AUTOTOOLS |
|---|---|---|---|---|
| **Exec** | 1.61 | 1.79 | 2.19 | 2.41 |
| **Utility** | 1.86 | 1.97 | 2.19 | 2.40 |

well-educated volunteers to evaluate 30 cases randomly sampled from our experiment benchmarks in Table 4. Details of human evaluation. Specifically, the annotators manually evaluate the task-solving trajectory step-by-step for Utility and Executability metrics using the ground truth solution as a reference. To guarantee annotation quality, we ask at least two annotators to evaluate the same example repeatedly. If there is a discrepancy between the two annotators (i.e., two annotators give a different score), we ask a third annotator to recheck it. The Kappa statistics for Executability and Tool utilization metrics are 0.70 and 0.69, which illustrates the agreement among annotators. Results of human evaluation. The results are shown in Table 9. We find that our method achieves the best in the Executability aspect with 0.21 absolute improvement compared with strong baselines, e.g., ToolLLM@3. We also observe that our method achieves higher performance on Utility. The reason for our superiority is that our framework enables the LLM to operate well-calibrated functions through programming, which is more executable compared with the manually designed workflow in previous work.

## A.3 A new benchmark – AUTOTOOLS-EVAL

Our AUTOTOOLS-EVAL benchmark is proposed to evaluate tool-use LLMs using more challenging tasks. Compared with the existing benchmark, our AUTOTOOLS-EVAL has the following advantages.

- **Long-term planning.** Most existing tool learning benchmarks are relatively simple, with each task being solved using 2 or 3 steps. However, real-world tasks often require complex workflows, such as `computing the rating scores for the top 10 newly released movies`. To reflect the tool learning capability of LLMs in realistic scenarios, each task in our AUTOTOOLS-EVAL benchmark is designed to involve at least 7 tool calls on average.
- **Connected reasoning.** Each task in our benchmark requires the model to interact with tools multiple times. To increase the challenge of the task, there is a strong interdependency among the tools, meaning that the argument of the current tool can only be extracted from the execution results of previous tools. This interdependent nature forces the models to connect information across all execution results of tools to solve a complex task, instead of simply making multiple calls without further reasoning.
- **Consistency and stability**: For high reproducibility, each task in our benchmark does not involve specific time, and the outputs of the tools are not time-varying.

We also compare our AUTOTOOLS-EVAL with existing benchmarks in Table 2.

*A.3.1 Details for benchmark construction.* Previous work like ToolBench [20] directly employs LLMs to generate datasets. However,

**Table 10: The statistics of our collected AutoTools-Eval benchmark, where we show the tool number and example number for each domain.**

| | Domain of the tools in our AutoTools-Eval | | | | Totally |
|---|---|---|---|---|---|
| | Food Recipe | Weather | Game | Movie | |
| Tasks | 64 | 50 | 50 | 60 | 224 |
| Tools | 22 | 11 | 20 | 54 | 107 |

it is proved to be less diverse or has unsolvable tasks [7, 49], raising concern about the scope and effectiveness of the evaluation. In this work, we adopt a bottom-up task collection approach driven by manual effort. Specifically, we employ 7 experts (*a.k.a.,* annotators) who work on NLP research to brainstorm tasks for different combinations of tools. Each expert is encouraged to integrate various tools to formulate a challenging task. Next, the experts need to manually solve these tasks with the assistance of candidate tools and annotate the ground truth solution, which includes the path of required tools and corresponding arguments for each tool calling. To establish a benchmark for highly consistent evaluations, we exclude any tasks where the solution varies over time. Specifically, a task is filtered out if the ground-truth solution path for the tool differs between two runs. Ultimately, we construct 227 examples across 107 tools from four domains. Table 11 shows an example of our collected benchmark. Compared with existing benchmarks which only list the required tools for each task, we further provide a ground truth solution for reference, including the required tools and corresponding arguments. Although the dataset is not large, each task in our benchmark is of high quality and represents the types of requests frequently made by users. The statistics of our benchmark are shown in Table 10.

*A.3.2 Strategy for quality improvement.* To ensure the quality of our constructed benchmark, we employ the following strategies.

- **Detailed annotator training.** We hold regular meetings to ensure that each expert has no questions about the annotation criteria. We also design pre-annotation tests, where each expert undergoes detailed training to familiarize themselves with our annotation task.
- **Cross-check for potential discrepancies.** To guarantee annotation quality, we ask at least two experts to annotate the same task repeatedly. If there is a discrepancy between the two experts, *i.e.,* two experts give different solutions for the same task, we ask a third expert to recheck it. We also filter the task with ambiguity to improve the reliability of our benchmark.
- **Periodic audits:** We conduct periodic audits of the annotations. These audits involved cross-checking a subset of annotated examples to verify compliance with the established criteria. We also held regular review meetings where annotation experts discussed challenging cases, ensuring a common understanding and application of the rules.

**Table 11: An example of our collected AUTOTOOLS-EVAL benchmark.**

| *Example of our AUTOTOOLS-EVAL benchmark (Food domain)* |
|---|

**Task:**
Please help me find a steak recipe and a pasta recipe. These recipes should have a carbohydrate content no higher than 80 grams per 100 grams, no lower than 5 grams per 100 grams. The protein content should be at least 5 grams per 100 grams for each recipe. Among them, which recipe requires fewer pieces of equipment, and how many ingredients does the recipe with fewer equipment contain?

**Base url for tool:**
https://spoonacular-recipe-food-nutrition-v1.p.rapidapi.com/

**Ground truth solution:**
1. GET /recipes/complexSearch

 - arguments: {"query": "steak", "minCarbs":5, "maxCarbs": 80, "minProtein": 5, "number": 1}
2. GET /recipes/complexSearch

 - arguments: {"query": "pasta", "minCarbs":5, "maxCarbs": 80, "minProtein": 5, "number": 1}
3. GET /recipes/recipe_id/equipmentWidget.json

 - arguments:{"recipe_id": 1094259}
4. GET /recipes/recipe_id/ingredientWidget.json

 - arguments: {"recipe_id": 1094259}
5. GET /recipes/recipe_id/equipmentWidget.json

 - arguments: {"recipe_id": 532245}
6. GET /recipes/recipe_id/ingredientWidget.json

 - arguments: {"recipe_id": 532245}

**Ground truth tools:**
1. GET /recipes/complexSearch
2. GET /recipes/{recipe_id}/equipmentWidget.json
3. GET /recipes/{recipe_id}/ingredientWidget.json
4. GET /recipes/{recipe_id}/equipmentWidget.json
5. GET /recipes/{recipe_id}/ingredientWidget.json
6. GET /recipes/{recipe_id}/similar

## A.4 Case Study

We conduct comprehensive case studies and find that our framework AutoTools is effective at coordinating various tools to solve complex tasks and our probing method can instruct the LLM to probe the input-output mechanism of tools, automatically synthesizing documentation. We provide the following cases to intuitively explain the details of our method.

**Example for Tool encapsulation.** Given a tools (e.g., a tool from Spotify dataset), the LLM automatically understand the key information in tool documentation, and transform abstract documentation into a well-encapsulated function. The function in our experiment is Python format, which can is directly callable.

The tool documentation of a tool named "GET_search" is shown below.

```
Tool: GET_search
Development documentation: {'method': 'GET', 'url': 'https://api.themoviedb.org/3/search/person', 'name':
 'GET_search_person', 'description': 'Search for people.', 'parameters': [{'name': 'query', 'in': 'query
', 'schema': {'type': 'string'}, 'description': 'Pass a text query to search. This value should be URI
encoded.', 'required': True}, {'name': 'page', 'in': 'query', 'schema': {'type': 'integer', 'default':
1}, 'description': 'Specify which page to query.'}, {'name': 'include_adult', 'in': 'query', 'schema': {'
type': 'boolean', 'default': False}, 'description': 'Choose whether to inlcude adult (pornography)
content in the results.'}, {'name': 'region', 'in': 'query', 'schema': {'type': 'string'}, 'description':
 'Specify a ISO 3166-1 code to filter release dates. Must be uppercase.'}], 'requestBody': None, 'example
': '{\n    "status_code": 7,\n    "status_message": "Invalid API key: You must be granted a valid key.",\
n    "success": false\n}', "responses": {"description": "", "content": {  "application/json": {"schema":
{"type": "object",
```

The LLM encapsulates the tool "GET_search" into a python function grounded with the above documentation, which is shown as below.

```python
import requests

def search_person(query: str, api_key: str, page: int = 1, include_adult: bool = False, region: str =
None) -> dict:
    """
    Call the GET_search_person API from TMDb to search for people.

    Parameters:
    query (str): The search term to find people (e.g., actor or director's name).
    api_key (str): Your TMDb API key.
    page (int, optional): The page number for pagination (default is 1).
    include_adult (bool, optional): Whether to include adult content (default is False).
    region (str, optional): ISO 3166-1 code to filter results by release dates in a specific region.

    Returns:
    dict: The JSON response from TMDb API with the search results.
    """
    url = 'https://api.themoviedb.org/3/search/person'

    params = {
        'query': query,
        'api_key': api_key,
        'page': page,
        'include_adult': include_adult
    }

    if region:
        params['region'] = region

    response = requests.get(url, params=params)

    if response.status_code == 200:
        return response.json()
    else:
        print(f"Error: {response.status_code}, {response.json().get('status_message')}")
        return {}
```

To verify the ecapsulated function, the LLM then generate a test instance using its commonsense knowledge, i.e., brainstorming a query related to the tool's description and required parameters. The generated instance by LLM is shown below. We have removed the user-specific "API-Key" for anonymization.

```
api_key = "YOUR_API_KEY"  # Replace with your valid TMDb API key
result = search_person(query="Christopher Nolan", api_key=api_key)
print(result)
```

**Example for LLM-generated program.** During the functional programming (Section 3.3), the LLM utilize encapsulated functions by generating executable programs. We show a specific example in our experiment, including query, LLM-generated program (i.e., solution) and the encapsulated functions used in the program. The query is shown below.

```
Query: Give me some cover images of movies directed by Christopher Nolan.
```

The generated program by gpt-3.5-turbo and the execution results.

```
headers = {"Authorization": f"Bearer eyJhbGciOiJIUzI1NiJ9.eyJhdWQiOiIwZ"}
# Search for Christopher Nolan
search_results = search_person("Christopher Nolan", headers)
if len(search_results) > 0:
    # Get the person ID of Christopher Nolan
    person_id = search_results[0]["id"]
    # Get movie credits for Christopher Nolan
    movie_credits = get_movie_credits(person_id, headers)
    # Get the cover images of movies directed by Christopher Nolan
    cover_images = get_director_movie_posters(movie_credits)
    print(cover_images)
else:
    print("Christopher Nolan not found.")
```

```
Output: ['/cXDFFv6nL0So.jpg', '/rQ884232vY6Zp.jpg', '/riVXh3EimGO5Itg.jpg']
```

We also show the used function in LLM-generated program, which is pre- encapsulated by the LLM itself in our encapsulation stage (Section 3.1).

```
def search_person(query: str, headers: dict) -> dict:
    """
    Call the GET_search_person API to search for a person by name.

    Parameters:
    - query (str): The name of the person to search for.
    - headers (dict): Authorization headers for the API request.

    Returns:
    - dict: {'id': 'int', 'keywords': [{'id': 'int', 'name': 'str'}]}
    """
    url = "https://api.themoviedb.org/3/search/person"
    params = { "query": query, "include_adult": False}
    response = requests.get(url=url, headers=headers, params=params)
    if response.status_code == 200:
        return response.json().get("results", [])
    else:
        print(f"Error: {response.status_code}")
        return {}

def get_movie_credits(person_id: int, headers: dict) -> dict:
    """
    Call the GET_person_person_id_movie_credits API to get the movie credits for a person.

    Parameters:
    - person_id (int): The ID of the person whose movie credits to retrieve.
    - headers (dict): Authorization headers for the API request.

    Returns:
    - dict: JSON response containing movie credits.
    """
    url = f"https://api.themoviedb.org/3/person/{person_id}/movie_credits"
    response = requests.get(url, headers=headers)
    if response.status_code == 200:
```

```python
        return response.json()
    else:
        print(f"Error: {response.status_code}")
        return {}

def get_director_movie_posters(movie_credits: dict) -> list:
    """
    Extracts the poster paths for movies directed by the person from their movie credits.

    Parameters:
    movie_credits (dict): JSON response containing the movie credits.

    Returns:
    list: A list of poster paths for the movies directed by the person.
    """
    cover_images = []
    for movie in movie_credits.get("crew", []):
        if movie.get("job") == "Director" and movie.get("poster_path"):
            cover_images.append(movie["poster_path"])
    return cover_images
```

## A.5 Experiment Instruction

We provide the instruction used in our experiment, including: (1) the instruction $\mathcal{I}_E$ to instruct the LLM to encapsulate a tool into a directly callable function; (2) the instruction $\mathcal{I}_{Func}$ to enable the LLM to integrate multiple pre-functions by generating a executable programs for task-solving, and (3) the instruction $\mathcal{I}_{Rel}$ to instruct the LLM to select relevant tools. The three instructions are shown below. We use the "" to indicate the query-specific input.

**Instruction for Encapsulation.**

```
I have a set of customized tools. Each API has a usage in its documentation to demonstrate how to access
it. According its usage, your task is to encapsulate them into well-structured Python functions, along
with a testing instance to demonstrate how to call these functions.

Your encapsulated functions should follow these key points:
1. Self-Contained: Each function must handle the API request (including making the call and processing
the response) and return the result. All required constants must be included within the function itself,
rather than relying on external variables.
2. Function Flexibility: Ensure the function is flexible enough to accept necessary parameters based on
the API's requirements.
3. Error Handling: The function should be robust enough to handle HTTP request errors. This includes
checking for unsuccessful status codes and faithfully returning the error message or exceptions
information.

Here is an output templte:```python
# import necessary lib

def API_NAME(PARAMs: type):
    """Description: add the description of the functionality
    Args:
    - PARAM 1 (type): explain the params
    - ...
    """
    # define the variable constants, like header or base url
    ...
    # request get/post/...
    ...
    # Error Handling for state code
    ...
    return response

# begin your testing instance
```

Here is the detailed development documentation of an API.
{t_doc}

Since you may need specific parameters, e.g., id, to call this API, I also provide you with some known
APIs to get the required value you need. For example, you should first obtain the requisite id or key
identifier of an entity and search the entity's information using the id.
{docs}

Your output:```
```

**Instruction for functional programming.**

```
Here are some real-world functions. You need to answer my question by writing Python programs to call a
series of functions and `print` the final answer. The functions is directly callable and has been loaded
in the Python execution environment.

{functions}

Read the provide functions carefully and integrate necessary functions to solve my query: {query}.
```

```
You need to provide Python code that can be executed directly. Please add the name of the used APIs in
Python comments for the attribution consideration. Try to write a correct Python program and avoid
grammar errors, e.g. `variable is not defined`.

Query: {query}
Your output:
```python
[Program]
```
```

**Instruction for selecting relevant tools.**

```
Here is an API along with its development documentation:
{doc}

This API has strong input-output dependencies with several other APIs listed below. Specifically, the
input parameters required for this API (e.g., id) can only be obtained from the output of one or more
APIs in the candidate list. To make a successful call to the given API, please help me select the related
 APIs that can provide the necessary input parameters. Here is a list of candidate APIs:
{api_list}

Please select the relevant APIs by listing their names in Python List format in one line (e.g., ["API 1",
 "API 2", ...]). You are encouraged to select any APIs you think might be useful.
Your output: [
```

Received 20 February 2007; revised 12 March 2009; accepted 5 June 2009

