# OpenReview forum: "Tool Learning in the Wild: Empowering Language Models as Automatic Tool Agents"
_ACM.org/TheWebConf/2025/Conference — WWW 2025 Poster_

### Official Review · Reviewer_D5Sf · 2024-11-17

**Novelty:** 4
**Technical Quality:** 5

**Review:**

This article proposes AutoTools, a unified framework that enhances LLM's ability to use tools. It uses LLM to encapsulate many tools into callable functions and directly calls these functions in downstream tasks to solve real-world problems.

advantage:
1. Standardized language and clear writing ideas.
2. The method is simple and effective, and the experiment is relatively sufficient.
3. Pay attention to and solve the problem that it is difficult to effectively manually label all tools in a large number of tool scenarios, and provide insights for the field.

disadvantage:
1. Is it deviating from the theme of the WWW conference and not clearly stating its relevance to the WWW?
2. The description of the method is not clear enough.

**Questions:**

1. Please explain the relevance of this article to WWW. Can you add some clearer explanations?

2. In the integration verification method, how do you generate diverse test cases to ensure the functions are correct? The test method here seems to have certain limitations, that is, a limited number of samples cannot prove that the tool is still effective in a variety of environments, and the limited number of samples generated cannot cover a variety of test boundaries.

3. In the test, did you first screen the tools and then use the prompt to guide LLM to select the tools? Notice that in the prompt template in the appendix, you input {functions} as the context to LLM. Did you input all the tools together to LLM or did you filter them? If you input all the tool descriptions together, is the context too long? If you filtered them, how did you filter them, and how to ensure fairness when compared with the baseline?

4. When the tool is not compiled and packaged successfully, will it be used later?

5. When a tool is considered correct because the test samples are not enough to find its errors, it will cause uncontrollable damage to downstream tasks. Is there a detection mechanism to deal with similar problems? This situation may cause limitations to the use of this method in real scenarios. How to deal with it.

6. When looking for dependencies between different tools, does the order of initialization have an impact? And how to deal with the long context when looking for dependencies caused by a large tool set.

**Reviewer Confidence:**

3: The reviewer is confident but not certain that the evaluation is correct

**Scope:**

2: The connection to the Web is incidental, e.g., use of Web data or API

---

### Official Review · Reviewer_b29a · 2024-11-20

**Novelty:** 4
**Technical Quality:** 5

**Review:**

This paper describes an approach to use LLMs for tool use, such that they solve a problem by reading the documentation of the tool on the web, generating functions, generate integration tests when the problem requires a sequence of tool uses, generates a program with interactions that creates a larger program to solve the problem by multiple function calling.  In addition, the paper describes a benchmark that was created to test the tool use.  Also, the work tests if smaller open LLMs can be fine tuned to help create programs that require the use of multiple tools.

At a broad level, the topic is interesting, and the results definitely seem promising compared to the state of the art.  The reservation at the broad level is the relevance of this work to the web.  One might argue that generating functions from documentation is accessing web resources, as is calling multiple web based API tools but I did not see anything in the motivation of the paper that addressed this point.  A second level of concern is the relationship to the track.  LLMs are a black box, the fact that they can read documentation and generate code, and integrate multiple tools is useful - but its unclear to what extent this has to do with the main thrust of the semantics track.

Drilling down a bit further, while the work was technically sound, I found the writing quite difficult to understand.  I add specific questions below but most are centered around the fact that the writing is unclear, in many sections.

**Questions:**

1. Line 318-322 state that "Since the LLM may hallucinate and miss necessary tool argument [52], we automatically compile the generated function into syntax tree [17] for syntax check. If any parameter name or type in the function signature does not exactly match the definitions in the tool documentation, the function is considered to fail the syntax check."  How do you have the gold standard for what the types of the functions should be?  Is this manually annotated or you know that because of synthetic data generation?
2.  Section 3.2 - for integration testing, where do you derive the unit tests?  Is that part of the gold standard as well?
3.  Section 4.2, lines 504-506 - what do those numbers indicate and how is it related to what is shown in Table 1.
4. Table 4 is confusing - the caption reads "The number of correctly encapsulated tools using our vanilla method and two variants on benchmarks (test set)."  - What is the vanilla method?  What are the two different variants? There are 4 benchmarks in the table, 5 models, with a --- separating gpt3.5 for some reason, and the Totally line isn't well explained.  I assume thats the total number in the benchmark because the text talks about percentages of success for different models but this is truly unnecessarily confusing.
5. Lines 616-617 same section - there is a discussion of traversal number and sampling number without a description of exactly what the procedure is to elicit encapsulation of functions, or how these parameters relate to it.
6.  The efficiency analysis is confusing too - exactly what do you count for consumed tokens?  Isn't reading documentation expensive to generate the code?  Isn't output tokens also expensive in any LLM model?

Minor
1. typo on 811 - I think you mean observe?

**Reviewer Confidence:**

3: The reviewer is confident but not certain that the evaluation is correct

**Scope:**

2: The connection to the Web is incidental, e.g., use of Web data or API

---

### Official Review · Reviewer_bxXN · 2024-12-01

**Novelty:** 6
**Technical Quality:** 6

**Review:**

Pros:
- The approach offers a novel method for automating tool learning for large language models (LLMs), significantly reducing the manual effort required for processing documentation and creating demonstrations.
- It presents a clear technical contribution with two main components: automatic tool encapsulation and programming-based tool integration.
- The empirical validation is strong, incorporating both existing benchmarks and a new challenging dataset (AutoTools-Eval).
- The approach practically enhances the capabilities of open-source LLMs through AutoTools-Learning, focusing on three targeted learning tasks.
- Thorough ablation studies illustrate the value of each component.
- There are clear efficiency gains regarding token consumption when compared to baseline models.

Cons:
- The discussion on failure modes and limitations of the approach could be more comprehensive, especially concerning complex tool interactions.
- The error analysis section (Table 7) would benefit from additional detailed examples.
- There is limited discussion about the computational costs associated with training AutoTools-Learning.
- The comparison to concurrent work in tool learning could be strengthened.

**Questions:**

1. How does the system manage API versioning and updates to documentation? Is there an automatic mechanism to detect when functions need to be re-encapsulated?

2. The paper demonstrates strong results with GPT-4; however, performance declines significantly with smaller models. Could you provide more details on the minimum model capabilities required for effective tool encapsulation?

3. During the integration verification step, how do you address situations where multiple valid tool combinations are possible? Is there a method to capture alternative valid paths?

**Reviewer Confidence:**

3: The reviewer is confident but not certain that the evaluation is correct

**Scope:**

4: The work is relevant to the Web and to the track, and is of broad interest to the community

---

### Official Review · Reviewer_KaCF · 2024-12-01

**Novelty:** 4
**Technical Quality:** 5

**Review:**

The paper presents an approach for LLM-based automatic generation of LLM-usable tools out of web APIs and their documentation.

On the **positive** side:

- the use of tools by LLMs is an important aspect

- the work is in most parts very clear and of high quality

- the results improve the state of the art


On the side of **improvements**:

- in the related work discussion, the comparison to related models such as ToolLLM is very brief, and for reviewers in the general area of "semantics and knowledge", the comparison is not completely clear. For example, it is said that "however, manually converting various tools into executable functions and carefully designing in-context examples for LLMs requires domain knowledge and experience". I am not an expert on ToolLLM and related papers, but my understanding of it is that it does not manually convert 16000+ API documentations. To make sure the reader appreciates the novelty of the approach, it would help to explicitly discuss this.

  It is clear from later discussion, e.g., Section 3 ("AutoTools (1) automatically transforms abstract tool documentation into a callable function library and (2) allows the LLM to flexibly integrate multiple tools with different usage using a unified programming language." and Section 2 "For example, ToolLLM [20] and Confucius [ 6] prompt LLMs to generate tool-use queries, then supplement them with chain-of-thought solutions that interleave tool names, tool arguments, and tool responses.") that the distinction is likely different than simply that previous approaches did not feature automation at all.

- I would see it helpful to explicitly discuss novelty and technical depth: Section 4.1 to a certain extent implicitly discusses novelty, but apart from that and even with that, it would really help the paper to state explicitly where novelty or technical contributions were necessary apart from the overall workflow - closely related to point 2 mentioned in the conclusion ("the limitations of handcrafted, ad-hoc control flows to integrate LLM generation with diverse tool-calling actions.")

- The paper could make the connection to the "semantis and knowledge" track more clear.

Detailed comments

- "they first pre-process toolset" is a bit unclear

- p1 / 96: the mentioning of "translate[source] -> target" is a bit unclear

- "the LLM are" singular/plural

- p2 / 204: "making it large to scale to massive toolset" is unclear

**Questions:**

I would like to have comments on the three points raised above, i.e. (numbered):

(1) in the related work discussion, the comparison to related models such as ToolLLM is very brief, and for reviewers in the general area of "semantics and knowledge", the comparison is not completely clear. For example, it is said that "however, manually converting various tools into executable functions and carefully designing in-context examples for LLMs requires domain knowledge and experience". I am not an expert on ToolLLM and related papers, but my understanding of it is that it does not manually convert 16000+ API documentations. To make sure the reader appreciates the novelty of the approach, it would help to explicitly discuss this.

  It is clear from later discussion, e.g., Section 3 ("AutoTools (1) automatically transforms abstract tool documentation into a callable function library and (2) allows the LLM to flexibly integrate multiple tools with different usage using a unified programming language." and Section 2 "For example, ToolLLM [20] and Confucius [ 6] prompt LLMs to generate tool-use queries, then supplement them with chain-of-thought solutions that interleave tool names, tool arguments, and tool responses.") that the distinction is likely different than simply that previous approaches did not feature automation at all.

(2) I would see it helpful to explicitly discuss novelty and technical depth: Section 4.1 to a certain extent implicitly discusses novelty, but apart from that and even with that, it would really help the paper to state explicitly where novelty or technical contributions were necessary apart from the overall workflow - closely related to point 2 mentioned in the conclusion ("the limitations of handcrafted, ad-hoc control flows to integrate LLM generation with diverse tool-calling actions.")

(3) The paper could make the connection to the "semantis and knowledge" track more clear.

In particular these points would help to support the "novelty" and "technical quality" ratings. Point (3) above would support the "scope" rating.

I have one further question on the availability of source code (for the "technical quality" rating):

(4) Possibly I did not find it, but did you share the source code as evidence for reviewing?

**Reviewer Confidence:**

3: The reviewer is confident but not certain that the evaluation is correct

**Scope:**

2: The connection to the Web is incidental, e.g., use of Web data or API

---

### Official Review · Reviewer_JL4C · 2024-12-01

**Novelty:** 3
**Technical Quality:** 5

**Review:**

**Summary:**
This manuscript introduces a method, named as AutoTools, to work as an LLM-based automatic tool agent for solving users' language-based tasks, which relies less on manual interactions compared to the previous methods. AutoTools transforms the tool documentations into callable programming functions with verified syntax  correctness, and integrates these functions to form more complicated programs as tools to answer users' requests. Besides, a labelled dataset is introduced that can be used to finetune the small-sized open source LLMs.

**Strong points:**

- S1: The workflow proposed is distinct from the previous methods, which has less dependency on human annotation and has a generic control flow. Overall, the experimental results demonstrate that the proposed method is effective, and the experiments are comprehensive.

- S2: AutoTools achieves better effectiveness while requiring less computation cost, making it more practical compared to other methods.


**Weak points:**

- W1: My main concern is regarding the methodology innovation aspect. The proposed workflow resembles more of an engineering solution that integrates several existing techniques, lacking a certain level of research novelty to some extent. Personally, I feel that the overall approach is heavily engineering-oriented. Both stages of AutoTools resemble utilizing the existing capabilities of LLMs to accomplish specific generation tasks by specific task-related inputs.

- W2: The construction process of the training dataset also relies on advanced LLMs, e.g., GPT-4o, which could limits the inherent capability of AutoTools. This would make AutoTools dependent on the LLM used to construct the dataset.

**Questions:**

- Q1: Overall, this paper is quite comprehensive, and I don't have many detailed questions, as they seem relatively unimportant. The main concern I have is regarding the innovation of AutoTools. Could you clarify what the innovative aspects are in terms of **methodology or techniques** of AutoTools? There's no need to elaborate on how the process guides the LLM to generate code based on the users' inputs anymore.

- Q2: As mentioned in Section 4.2, the training data is constructed under the guidance of GPT-4o. Does it mean that the performance upper bound of the small-LLM-based AutoTools is equivalent to that of GPT-4o?

- Q3: What is the difference between AutoTools and LLM-based code generation methods? In my view, AutoTools is more like transforming the tool agent into a more complex, and multi-step code generation problem.

**Reviewer Confidence:**

3: The reviewer is confident but not certain that the evaluation is correct

**Scope:**

3: The work is somewhat relevant to the Web and to the track, and is of narrow interest to a sub-community